# Long-Term Evolution of Activities of Daily Life (ADLs) in Critically Ill COVID-19 Patients, a Case Series

**DOI:** 10.3390/healthcare11050650

**Published:** 2023-02-23

**Authors:** Samuele Ceruti, Andrea Glotta, Maira Biggiogero, Martino Marzano, Giovanni Bona, Marco Previsdomini, Andrea Saporito, Xavier Capdevila

**Affiliations:** 1Department of Critical Care, Clinica Luganese Moncucco, 6900 Lugano, Switzerland; 2Clinical Research Unit, Clinica Luganese Moncucco, 6900 Lugano, Switzerland; 3Department of Internal Medicine, Clinica Luganese Moncucco, 6900 Lugano, Switzerland; 4Department of Intensive Care Medicine, Ente Ospedaliero Cantonale, 6500 Bellinzona, Switzerland; 5Service of Anesthesiology, Ente Ospedaliero Cantonale, 6500 Bellinzona, Switzerland; 6Department of Anesthesia and Intensive Care, Centre Hospitalier Universitaire de Montpellier, 34000 Montpellier, France

**Keywords:** SARS-CoV-2, activities of daily life, post-acute COVID-19 syndrome, PICS, Barthel index, Karnofsky Performance Status, functional status

## Abstract

Background: The most common long-term symptoms of critically ill COVID-19 patients are fatigue, dyspnea and mental confusion. Adequate monitoring of long-term morbidity, mainly analyzing the activities of daily life (ADLs), allows better patient management after hospital discharge. The aim was to report long-term ADL evolution in critically ill COVID-19 patients admitted to a COVID-19 center in Lugano (Switzerland). Methods: A retrospective analysis on consecutive patients discharged alive from ICU with COVID-19 ARDS was performed based on a follow-up one year after hospital discharge; ADLs were assessed through the Barthel index (BI) and the Karnofsky Performance Status (KPS) scale. The primary objective was to assess differences in ADLs at hospital discharge (*acute ADLs*) and one-year follow-up (chronic ADLs). The secondary objective was to explore any correlations between ADLs and multiple measures at admission and during the ICU stay. Results: A total of 38 consecutive patients were admitted to the ICU; a *t*-test analysis between acute and *chronic ADLs* through BI showed a significant improvement at one year post discharge (t = −5.211, *p* < 0.0001); similarly, every single task of BI showed the same results (*p* < 0.0001 for each task of BI). The mean KPS was 86.47 (SD 20.9) at hospital discharge and 99.6 at 1 year post discharge (*p* = 0.02). Thirteen (34%) patients deceased during the first 28 days in the ICU; no patient died after hospital discharge. Conclusions: Based on BI and KPS, patients reached complete functional recovery of ADLs one year after critical COVID-19.

## 1. Introduction

The SARS-CoV-2 virus emerged in December 2019 in Wuhan, China, and spread rapidly across the globe [1]; on 20 February 2020, the first Swiss patient with SARS-CoV-2-related pneumonia was accepted in our center in Lugano [2]. In March 2020, the World Health Organization (WHO) declared a global pandemic due to this new coronavirus (COVID-19) [3]; in the following two years, John Hopkins University’s Coronavirus Resource Center has recorded more than 630 million cases and more than 6 million deaths [4].

Patients who develop COVID-19-related acute respiratory distress syndrome (ARDS) have a high mortality rate (26–55%) [5,6,7,8,9], but apparently, those who survive the acute phase have a good prognosis with a 6-month mortality rate of 1.3–3% [9,10]. However, concerns remain that many survivors may suffer from severe long-term physical and psychological sequelae, both in terms of physical disability and post-traumatic stress disorder (PTSD) [11]. COVID-19 survivors report numerous physical symptoms and psychosocial disorders even one year after the acute illness [12,13,14,15,16,17,18,19,20,21,22]. Three months after acute COVID-19, 50 to 70% of patients have signs of fibrosis on CT [12,13,14] and, proportionally to the severity of the acute disease [15,16], 25–50% have lung function alterations, mainly concerning reduced diffusion capacity for carbon monoxide (D_LCO_), which may persist unchanged at 12 months [17,18,19]. In the subpopulation of critically ill patients admitted to the ICU for COVID-19 pneumonia, radiological alteration, pulmonary restriction and D_LCO_ impairment are even more widespread and pronounced [9,19,20,21,22,23,24] and, being associated with long-term injuries [24,25], result in a reduced 6 min walking distance up to 12 months after hospital discharge [9].

Following the acute phase, emotional and psychological distress may persist for an extended period, further contributing to multifactorial disabilities requiring continuous care and multidisciplinary rehabilitation management [26,27,28]. A recent meta-analysis reported a significant incidence of post-traumatic stress disorder (39%), symptoms of depression (33%) and anxiety (30%) affecting survivors of the most severe forms of COVID-19 six months after hospital discharge [29]. Careful evaluation of long-term morbidity outcomes in critically ill COVID-19 patients is essential to optimize patient management after hospital discharge [30,31]; an essential aspect regarding physical sequelae concerns the ability to regain independence in *activities of daily life* (ADLs).

The aim of this study was to describe the long-term physical consequences in critically ill COVID-19 patients, identifying long-term sequelae in patients’ ADLs through a follow-up analysis of the Barthel index (BI) and Karnofsky Performance Status (KPS) scale one year after hospital discharge.

## 2. Materials and Methods

A retrospective analysis of consecutive critically ill COVID-19 patients discharged alive from the ICU in our COVID-19 center during the first pandemic wave was performed, from 16 March to 10 April 2020. Patients transferred to other ICUs were excluded from the analysis. We included patients with ARDS due to a SARS-CoV-2 infection confirmed by a positive result from real-time reverse transcriptase-polymerase chain reaction (RT-PCR) on nasal or pharyngeal swabs [32]. Clinical data regarding the acute phase, defined as the hospitalization time in the ICU and acute medicine department, from electronic health records were retrieved. We collected the following demographic and clinical characteristics: age, sex, body mass index (BMI), comorbidities (arterial hypertension (HTA), diabetes mellitus (DM), obstructive sleep apnea syndrome (OSAS), chronic obstructive pulmonary disease (COPD) and ischemic heart disease (IHD)), pulmonary embolism (VTE), days of symptoms before hospital admission, ICU length of stay (ICU LOS), Simplified Acute Physiology Score II (SAPS) during the first 24 h, admitting Sequential Organ Failure Assessment (SOFA), days of mechanical ventilation, pronation sessions, presence of tracheostomy and development of ventilator-associated pneumonia (VAP). Further, we collected the following laboratory test results: ASAT, ALAT, leucocytes, lymphocytes, lactates (all at admission in the ICU), minimum platelet level and maximum total bilirubin, CK, CRP, LDH, ferritin and creatinine (during the entire ICU stay).

A follow-up evaluation one year after hospital discharge was performed by contacting patients by telephone to gather information on survival and their performance status.

### 2.1. Performance Evaluation

During the interviews, the Italian-validated versions of two widely used international scales, the BI [33,34,35] and the KPS scale [36,37], were administered (Appendix A).

The BI measures ten essential self-care and physical dependency aspects, rating each ADL’s element on a semiquantitative scale with high inter-rater and test-retest reliability [38]. A score of 100 denotes normality, and lower scores indicate increasing disability. Notably, there is a direct correlation between the ability to carry out these everyday activities and the degree of autonomy necessary to live at home after hospital discharge [33]. Single ADLs, such as getting out of bed, going to the toilet, dressing and eating, can be used as markers of an individual’s functional status during standard and serial ADL screening to detect the presence and the degree of a specific disability [39].

The KPS was primarily developed to assess a patient’s ability to survive chemotherapy [40,41,42,43]. However, it is helpful for assessing functional impairment by measuring a patient’s general performance status or ability to carry out ADLs [44]. 

### 2.2. Outcomes

The primary objective was to evaluate the difference in ADLs at hospital discharge (*acute ADLs*) and 12 months later (*chronic ADLs*) by administering the BI and KPS. We looked for correlations between acute/chronic ADLs and clinical and biological characteristics observed during the ICU stay as a secondary objective.

### 2.3. Statistical Analysis

Descriptive statistics were used to summarize the clinical data. Data are presented as mean (SD) or median (IQR) for continuous variables according to the data distribution and as absolute numbers (percentage) for categorical variables. The data distribution was verified by the Kolmogorov–Smirnov and Shapiro–Wilk tests (Appendix A). Differences between continuous variables by the paired t-test or the Mann–Whitney test for independent groups requiring a nonparametric analysis were investigated. Clinical evolution over time was compared using the paired t-test or the nonparametric Wilcoxon test depending on data distribution, and we analyzed the relationships between continuous variables by linear regression. All intervals of confidence (CI) were established at 95%. The type 1 error rate was 0.05. Statistical significance was considered with a p value less than 0.05. Statistical data analysis was performed using the SPSS.26 package (SPSS Inc., Armonk, NY, USA).

## 3. Results

Among 38 critically ill COVID-19 patients admitted with ARDS, 12 (31.6%) died in the ICU and 1 (2.6%) died in the hospital; none of the 25 survivors included in our analysis died during the follow-up year (Figure 1). 

The mean age was 59 years (SD 12), 21 (84%) were men and the mean BMI was 29 kg/m^2^ (SD 5); 11 (44%) patients had arterial hypertension, 8 (32%) had diabetes, 3 (12%) had OSAS, 1 (4%) had COPD and 1 (4%) patient presented pulmonary embolism during the SARS-CoV-2 infection. The survivors’ mean ICU length of stay (LOS) was 12 days (SD 8). The mean duration of mechanical ventilation (MV) was 11.8 (SD 9) days; 20 (80%) patients received invasive MV, 4 (16%) patients received a tracheostomy and 4 (16%) patients required continuous renal replacement therapy (CRRT). At ICU admission, no patients required vasopressors. Demographic, clinical and laboratory characteristics are shown in Table 1.

### 3.1. Primary Outcome

The median BI was 75 (IQR 55–97.5, min/max 5–100) at hospital discharge (*acute ADLs*) and 100 (IQR 100–100; diff = 25, Z = −3.823, *p* < 0.0001) at one year post discharge (*chronic ADLs*); chronic ADLs showed a complete recovery in all analyzed activities (Figure 2).

Data regarding the single BI activities at hospital discharge (*acute ADLs*) and at one-year follow-up (*chronic ADLs*) were compared and show significant differences for several items (Table 2).

The mean KPS value at hospital discharge was 86.47 (SD 20.9, min/max 20–100) and 99.6 (SD 2.0, min/max 90–100, t = −2.583, dF 24, *p* = 0.02) at one year post discharge, with nearly complete recovery in working activities (98.8, SD 6.0, min/max 70–100), ADLs measured by KPS (99.6, SD 2.0, min/min 90–100) and personal care (99.6, SD 2.0, min/max 90–100).

### 3.2. Secondary Outcomes

No significant correlations between the BI at discharge and demographic, biological and ICU-specific data were found (Table 3) except for a slight trend between the SOFA score at ICU admission and acute BI (r^2^ = 0.2).

A slight reverse correlation between the BI at hospital discharge and ICU LOS (r^2^ = 0.08), MV days (r^2^ = 0.117) and the SOFA score at ICU admission (r^2^ = 0.2, Figure 3) was observed. 

Patients with OSAS (t = −0.833, dF 2.207, *p* = 0.485) and those treated with invasive MV (t = −2.173, dF 13.764, *p* = 0.048) and with nosocomial infections (t = −0.383, dF 4.972, *p* = 0.718) presented a slightly different BI at hospital discharge than the rest of the cohort, without any significant differences. A similar reverse correlation between KPS at hospital discharge and ICU LOS was found (r^2^ = 0.661, *p* = 0.02), and simple correlation trends between KPS and MV days (r^2^ = 0.536, *p* = 0.385) and between KPS at hospital discharge and age were similarly encountered (r^2^ = 0.185, *p* = 0.139, Figure 4).

Further, no correlations between any single *acute ADL* element of the BI and clinical, biological or intra-ICU data were found (Table 4). 

In particular, there was no correlation between feeding and invasive MV/tracheostomy (Z = −1.065, *p* = 0.287 and Z = −0.639, *p* = 0.523, respectively), between walking and invasive MV/tracheostomy (Z = −1.432, *p* = 0.152 and Z = −0.329, *p* = 0.742, respectively) or walking and ICU LOS/MV days (r^2^ = 0.038, *p* = 0.739 and r^2^ = 0.018, *p* = 0.536, respectively).

## 4. Discussion

Our data showed that critically ill COVID-19 patients discharged from the hospital presented significant improvements within the first year of follow-up, fully recovering ADLs and KPS functional status. 

With a mean BI of 75 and a mean KPS of 86 at hospital discharge, critically ill COVID-19 patients presented a moderate degree of ADL dependency at hospital discharge and could perform regular activities with some effort despite some signs or symptoms of disease. One year later, our entire cohort achieved a BI of 100, meaning complete functional autonomy, and a KPS of 99.6, which indicates the absence of complaints and the ability to carry on regular activities and work. These results are encouraging because they showed that COVID-19-related ARDS survivors can cope with the massive catabolic state inherent to the acute critical illness and the numerous symptoms and limitations reported by patients up to 12 months after acute COVID-19 illness, called the post-COVID-19 syndrome [9,17,19,45,46]. 

In a Spanish multicentric cohort of 113 patients, the mean BI was 99 at a median of 240 days from the first positive PCR test for SARS-CoV-2 despite 80.5% having at least one residual symptom [9]. One of the most frequent symptoms was fatigue, which affects about 60% of patients one year after discharge [19,47]. It may persist as a direct consequence of some level of immune activation with or without persistent viral infection [48,49,50,51,52,53,54], long-term lung tissue damage, lasting neurological complications [55,56], myocardial injury [57,58] and other extrapulmonary involvement [59]. Impaired muscle function and deconditioning may also explain the compromised functional ability, impacting the 6 min walking distance. Notably, virtually all ARDS patients exhibited severe muscle waste and weakness in the acute phase, and only 70% returned to their baseline weight by one year [25]. Nine months after SARS-CoV-2-induced ARDS, about half of the patients completed less than 80% of the theoretical reference distance in the 6 min walk distance test (6MWDT) [9]. In two other studies, the 6MWDT showed significant recovery between 3 and 12 months in most ICU survivors after SARS-CoV-2 pneumonia but remained below the predicted value for 20–25% of patients with no association with D_LCO_ alterations. The functional improvement occurred despite significant radiological lung parenchyma alterations (reticulations, traction bronchiectasis, honeycombing, ground-glass opacities and emphysema) persisting in most patients (80–95%) 12 months after COVID-19 related ARDS [47,60]. 

Additionally, 40–45% of patients report some degree of breathlessness, mainly modified Medical Research Council Dyspnea scale (mMRC) grades 1 and 2, one year after acute SARS-CoV-2 pneumonia, in some cases with a worsening trend compared to the 6-month follow-up [9,19,47,61,62]. At this time point, reduced D_LCO_ and pulmonary restriction still affect approximately 50–60% and 7–30% of patients and are proportional to the severity of lung failure expressed as the level of respiratory support during the acute SARS-CoV-2 infection [19,47]. 

According to reports on COVID-19-related ARDS survivors, the health-related quality of life (HRQoL) was worse than in the general population [9,60], which is also consistent with previous studies among patients affected by ARDS of different etiologies [25,63]. Mean Short-Form 36 (SF36) scores were significantly worse than the general population for each of the eight dimensions in a Spanish multicentric cohort eight months post discharge [9]. In a second French monocentric observational study, SF36’s emotional role domain normalized three months after hospital discharge and the physical role within 12 months, but the other six domains continued to show reduced scores [60]. Psychological and emotional dysfunction is known to persist for up to five years after ICU discharge [64]. Accordingly, many patients develop anxiety disorders (30%), depression (33%) and post-traumatic stress disorder (PTSD) symptoms (39%) [29], potentially aggravated by the past pandemic period.

Regardless of the primary illness, survivors of an extended ICU stay may experience medium- and long-term morbidity related to critical illness, the necessary support and the environment. This condition, which may include new or worsening cognitive, psychiatric and physical impairment, is now recognized as *post-intensive care syndrome* (PICS) [65,66,67,68,69]. Therefore, survivors of critical COVID-19 may experience a range of sequelae related to their critical condition (i.e., PICS), the SARS-CoV-2 infection (i.e., post-acute COVID-19 syndrome (PACS)) or both. In the present study, no patients reported any specific or persistent symptoms 1 year after hospital discharge.

This study is part of the constantly growing group of scientific evidence reporting the medium-long term clinical conditions of critical COVID-19 patients, not only confirming that the post-discharge mortality rate is confirmed to be low but also, above all, underlining how functional recovery—despite potential residual symptoms—appears to be almost complete, with the possibility of having a quality of life completely similar to the situation prior to admission. We recently showed that critically ill COVID-19 patients could recover certain physical functions (swallowing) significantly faster than different critically ill patients [70]. This information appears even more useful observing the correlation between the BI at hospital discharge and the SOFA score at ICU admission, suggesting that even for patients initially considered most critical, the chances of long-term recovery are optimal once they are discharged from the ICU. Precisely in line with this peculiarity of COVID-19 patients, Biehl et al. [71] analyzed non-COVID-19 critically ill patients with/without ARDS at 6 months using the Barthel index, finding no significant differences between baseline and 6-month ADLs. These findings suggested that critically ill COVID-19 patients have the potential to recover better after the acute phase of damage than critical patients with ARDS of another nature for a number of causes related to the disease and the characteristics of patients discharged alive from the ICU, which will certainly be the subject of future studies.

This study presented some limitations. Firstly, it was a single-center retrospective study enrolling a fairly small number of patients; although our results are in line with other groups [9,62], further confirmatory studies are needed. Second, the follow-up assessment was performed over the phone and not face-to-face in the clinic. The information collected was based on the patient’s self-assessment and not on a direct medical evaluation, although it is important to note that the BI and the KPS can be administered reliably via telephone conversation [72]. Again, we do not have any homogenous information about the rehabilitation programs the patients participated in, as they were transferred to different institutions with patient-specific nonstandardized programs. Finally, we did not look for the presence of psychiatric or psychological disorders either in the acute or in the chronic phase, although they can affect ADLs [73,74] both positively and negatively; the Barthel index is designed to monitor pure functional autonomy for ADLs and does not cover cognitive decline, which of course could affect the global functional result.

## 5. Conclusions

Critically ill COVID-19 patients showed complete recovery of ADLs and performance status one year after acute illness. Despite PICS and PACS, this long-term perspective justifies prolonged hospitalization in the ICU and the use of invasive and aggressive techniques to overcome the most acute and dangerous phase.

## Figures and Tables

**Figure 1 healthcare-11-00650-f001:**
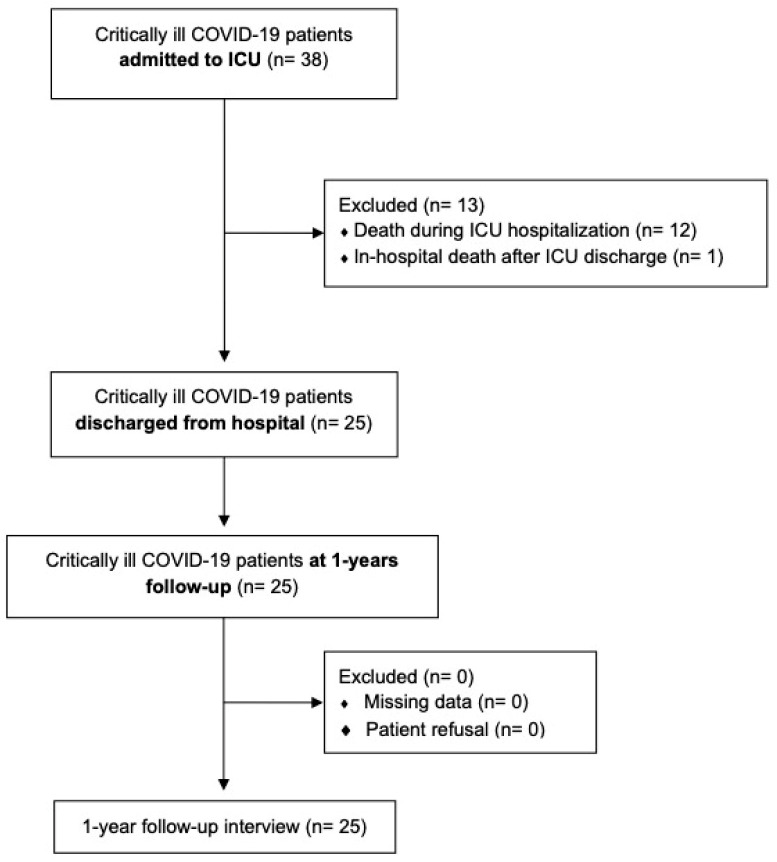
Patients’ distribution flowchart. Critically ill COVID-19 patients’ distribution according to temporal evolution. No patient died during the follow-up period.

**Figure 2 healthcare-11-00650-f002:**
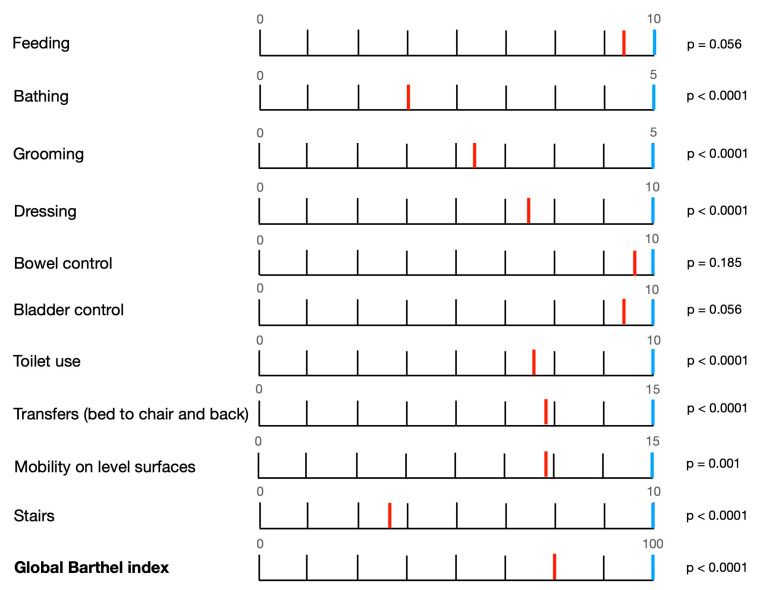
Temporal evolution of ADLs according to Barthel index. Temporal evolution of single activity of daily life (ADLs) comparing *acute ADLs* (red bar) and *chronic ADLs* (blue bar) with their specific statistical significance.

**Figure 3 healthcare-11-00650-f003:**
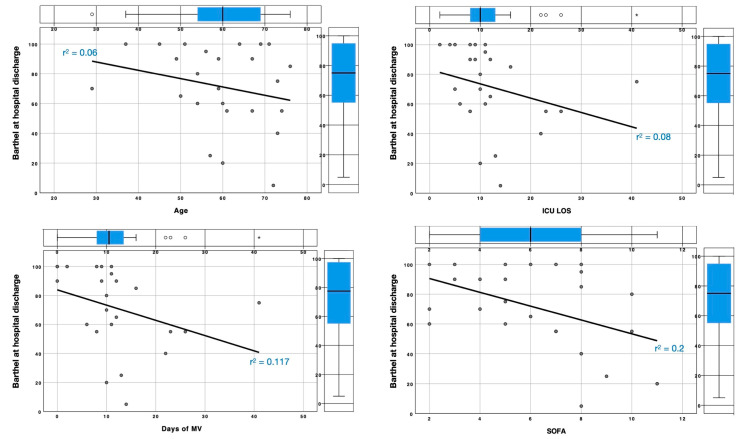
Linear regression of acute Barthel index. Linear regressions of Barthel index at hospital discharge compared to age, ICU LOS, days of MV and admitting SOFA score. ICU LOS = intensive care unit length of stay, MV = mechanical ventilation and SOFA = Sequential Organ Failure Assessment.

**Figure 4 healthcare-11-00650-f004:**
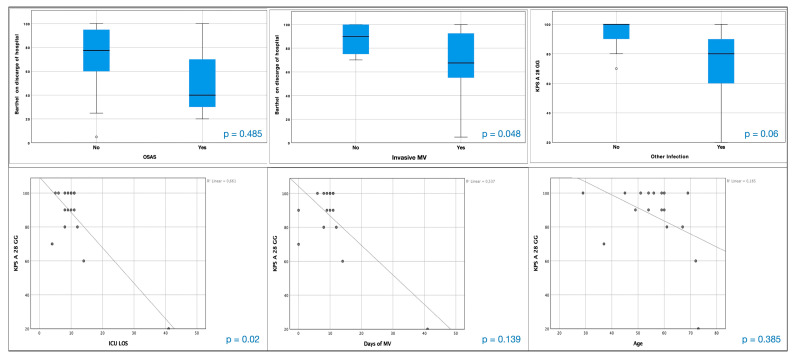
Analysis of boxplot of Barthel index. Boxplot stratification of Barthel index at hospital discharge comparing patients with OSAS, invasive MV and the presence of other infections. OSAS = obstructive sleep apnea syndrome and MV = mechanical ventilation.

**Table 1 healthcare-11-00650-t001:** Patients’ baseline characteristics.

	Unit	Values
**Demographics data**		
Age	yrs	59 ± 12 (29–76)
Male sex	n (%)	21 (84%)
BMI	Kg/m^2^	29 ± 5 (20.1–40.1)
HTA	n (%)	11 (44%)
DM	n (%)	8 (32%)
OSAS	n (%)	3 (12%)
COPD	n (%)	1 (4%)
Pulmonary embolism	n (%)	1 (4%)
**ICU Data**		
SAPS II (first 24 h in ICU)		39 ± 15 (13–70)
SOFA *		6 ± 2 (2–11)
NEMS		33 ± 9 (18–42)
Temperature *	°C	37.2 ± 0.9 (35.9–39.0)
Systolic arterial blood pressure *	mmHg	133 ± 21 (100–180)
Diastolic arterial blood pressure *	mmHg	70 ± 16 (50–110)
Heart rate *	bpm	91 ± 25 (55–160)
Oro-tracheal intubation	n (%)	20 (80%)
Tracheostomy	n (%)	4 (16%)
CRRT	n (%)	4 (16%)
Ventilation-associated pneumonia	n (%)	4 (16%)
Other ICU infections	n (%)	5 (20%)
ICU LOS	days	12 ± 8 (2–41)
MV days	days	11.8 ± 9 (0–41)
**Laboratory Data**		
White cells *	G/L	8.2 ± 3.5 (3.5–14.9)
Lymphocytes *	G/L	0.9 ± 0.7 (0.2–4.1)
Lactate *	mmol/L	1.1 ± 0.4 (0.5–2.1)
ASAT *	U/L	66 ± 30 (22–131)
ALAT *	U/L	53 ± 26 (25–123)
CRP max	mg/L	241 ± 128 (57–534)
LDH max	U/L	671 ± 390 (184–2291)
Ferritin max	ng/mL	2354 ± 2307 (455–11,000)
Creatinine max	μmol/L	124 ± 117 (50–521)
Thrombocytes min	G/L	245 ± 84 (111–458)
Bilirubin max	μmol/L	13.2 ± 12.5 (3.8–52)
CK max	U/L	424 ± 386 (33–1680)

Patients’ characteristics during the ICU stay regarding clinical, laboratory and intra-ICU data. Data on SAPS II, SOFA, NEMS, systolic and diastolic arterial blood pressure, heart rate and temperature were reported from ICU admission; *: at ICU admission. Continuous measurements are presented as mean ± SD (min–max) and otherwise as median (25th–75th) if they are not normally distributed. Categorical variables are reported as counts and percentages. No patients were on vasopressor at ICU admission. CRRT = continuous renal replacement therapies, ICU = intensive care unit, ICU LOS = intensive care unit length of stay, MV = mechanical ventilation, ASAT = aspartate-aminotransferase, ALAT = alanine-aminotransferase, CRP = C-reactive protein, LDH = Lactate dehydrogenase and CK = creatine kinase.

**Table 2 healthcare-11-00650-t002:** Acute and chronic ADLs (Barthel index).

	Acute ADLs	Chronic ADLs	t	dF	*p* Value
Feeding	8.8 ± 2.9	10.0 ± 0			0.056
Bathing	2.0 ± 2.5	5.0 ± 0	−6.0	24	<0.0001 *
Grooming	2.6 ± 2.5	5.0 ± 0	−4.707	24	<0.0001 *
Dressing	6.8 ± 3.8	10.0 ± 0	−4.226	24	<0.0001 *
Bowel control	9.4 ± 2.2	10.0 ± 0			0.185
Bladder control	8.8 ± 3	10.0 ± 0			0.056
Toilet use	7.0 ± 3.5	10.0 ± 0	−4.243	24	<0.0001 *
Transfers (bed to chair and back)	11.2 ± 4.6	15 ± 0	−4.106	24	<0.0001 *
Mobility on level surfaces	11.4 ± 4.9	15 ± 0	−3.674	24	0.001 *
Stairs	3.4 ± 4.2	10.0 ± 0	−7.742	24	<0.0001 *
Global Barthel index	75 (55–97.5)	100 (100–100)	−5.211	24	<0.0001 *

Acute and chronic ADL evaluation performed by Barthel index at hospital discharge (acute Barthel) and at follow-up of 1 year (chronic ADLs). Data are reported as mean (SD) or median (IQR) depending on data distribution according to Kolmogorov–Smirnov test. *: statistically significant.

**Table 3 healthcare-11-00650-t003:** Correlation analysis with acute activities of daily life (ADLs) evaluated by Barthel index.

	Correlation	*p* Value
**Demographics Data**		
Age	−0.219	0.158
BMI	−0.167	0.223
AHT	−0.497	0.619
DM	−0.294	0.769
OSAS	−0.886	0.376
COPD	−0.979	0.327
Pulmonary embolism	−0.146	0.884
**ICU Data**		
SAPS II (first 24 h in ICU)	0.070	0.376
SOFA *	−0.473	0.01 *
Systolic arterial blood pressure *	−0.03	0.446
Diastolic arterial blood pressure *	0.004	0.492
Heart rate *	0.04	0.428
Temperature *	−0.031	0.444
Oro-tracheal intubation	−1.473	0.141
Tracheostomy	−0.935	0.35
CRRT	−0.187	0.852
Ventilation-associated pneumonia	−0.336	0.737
Other ICU infections	−0.206	0.837
Days of symptoms before admission	0.235	0.14
ICU LOS	−0.275	0.102
MV days	−0.322	0.067
Pronation sessions	0.071	0.373
**Laboratory Data**		
White cells	−0.353	0.05 *
Lymphocytes	−0.042	0.425
Lactate	−0.322	0.067
ASAT	0.004	0.493
ALAT	0.119	0.294
CRP max	0.033	0.441
LDH max	−0.238	0.137
Ferritin max	0.102	0.322
Creatinine max	0.163	0.229
Thrombocytes min	0.213	0.165
Bilirubin max	−0.184	0.2
CK max	−0.142	0.258

Correlation analysis performed between *acute ADLs* (BI) at hospital discharge and clinical, biological and intra-ICU variables. The correlation analysis was performed according to data distribution through Pearson correlation, regression correlation or Mann–Whitney test. *: at ICU admission. BMI = body mass index, AHT = arterial hypertension, DM = diabetes mellitus, OSAS = obstructive sleep apnea syndrome, COPD = chronic obstructive pulmonary disease, SAPS II = Simplified Acute Physiology Score, SOFA = Sequential Organ Failure Assessment, NEMS = Nine Equivalents of Nursing Manpower Use Score, CRRT = continuous renal replacement therapy, ICU LOS = intensive care unit length of stay, MV = mechanical ventilation, ASAT = aspartate-aminotransferase, ALAT = alanine-aminotransferase, CRP = C-reactive protein, LDH = Lactate dehydrogenase and CK = creatine kinase.

**Table 4 healthcare-11-00650-t004:** Correlation analysis of acute activities of daily living (ADLs).

	Sex	HTA	DM	OSAS	COPD	PE	IMV	Trach	CRRT	VAP	Inf
Feeding	−0.407 0.684	−0.429 0.668	−0.228 0.819	−0.983 0.326	−0.435 0.664	−0.445 0.656	−1.065 0.287	−0.639 0.523	−0.639 0.523	−0.639 0.523	0.373 0.709
Bathing	−0.436 0.663	−0.322 0.747	−0.171 0.864	−0.246 0.806	−0.816 0.414	−0.845 0.398	−1.0 0.317	−1.746 0.081	−0.436 0.663	−0.655 0.513	−1.0 0.317
Grooming	−0.984 0.325	−0.569 0.569	−0.975 0.329	−0.676 0.499	−1.041 0.298	−1.0 0.317	−1.373 0.17	−1.155 0.248	−0.086 0.932	−0.086 0.932	−0.588 0.588
Dressing	−0.245 0.806	−1.538 0.124	−0.385 0.7	−1.382 0.167	−0.688 0.492	−0.881 0.378	−1.46 0.144	−1.102 0.27	−0.245 0.806	−0.245 0.806	−0.561 0.575
Bowel control	−0.63 0.529	−1.628 0.103	−0.619 0.536	−1.599 0.11	−0.295 0.768	−0.301 0.763	−0.722 0.47	−1.417 0.156	−1.26 0.208	−1.26 0.208	−1.155 0.248
Bladder control	−0.407 0.684	−1.416 0.157	−1.005 0.315	−0.983 0.326	−0.435 0.664	−0.445 0.656	−0.16 0.873	−1.975 0.048 *	−0.93 0.353	−0.93 0.353	−1.597 0.11
Toilet use	−0.164 0.869	−0.729 0.466	−0.129 0.897	−0.928 0.353	−0.769 0.442	−0.717 0.473	−1.432 0.152	−0.329 0.742	−0.329 0.742	−0.329 0.742	−0.151 0.88
Transfers	−0.405 0.686	−0.777 0.437	−0.413 0.679	−1.096 0.273	−1.363 0.173	−0.47 0.639	−1.521 0.128	−0.04 0.968	−0.486 0.627	−0.486 0.627	−0.408 0.683
Mobility on level surfaces	−0.164 0.869	−0.729 0.466	−0.129 0.897	−0.928 0.353	−0.769 0.442	−0.717 0.473	−1.432 0.152	−0.329 0.742	−0.329 0.742	−0.329 0.742	−0.151 0.88
Stairs	−1.033 0.301	−1.13 0.259	−0.325 0.745	−0.14 0.889	−0.851 0.395	−0.816 0.414	−0.909 0.363	−0.248 0.804	−0.207 0.836	−0.207 0.836	−0.493 0.622

Correlation analysis between ADLs stratified according to each single BI task, demographics and clinical and intra-ICU occurrence. Linear regression analysis with continuous variables did not identify any other correlations. Data are reported as z-test and *p* value. *: statistically significant. HTA = arterial hypertension, DM = diabetes mellitus, PE = pulmonary embolism, IMV = invasive mechanical ventilation, Trach = tracheostomy, CRRT = continuous renal replacement therapy, VAP = ventilation-associated pneumonia and Inf = other intra-ICU infectious diseases.

## Data Availability

The data presented in this study are available on request from the corresponding author.

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
