# Peer review of "Long-Term Evolution of Activities of Daily Life (ADLs) in Critically Ill COVID-19 Patients, a Case Series"

_healthcare, 2023, doi:10.3390/healthcare11050650_

Round 1
Reviewer 1 Report
General comments
The manuscript (MS) aimed to report long-term ADL evolution in critically ill COVID-19 patients, one year after hospital discharge. The MS is well presented and well written. I have only some comments which need to be addressed.
Some minor English editing is needed which I will not address in the review.
Specific comments
Introduction
1. Line 67 – “daily life activities (ADL)” - Considering the authors used the acronym “ADL” I believe it will be better to write “activities of daily life”.
Results
2. Line 128, Figure 1 - Something is missing in the flow chart (e.g. boxes, arrows, etc.).
3. Lines 130-131 - Is this the title of a subsection of the Results? The size of the letters is similar to the one used in the legend of the Figure 1.
4. Line 141, Table 1 – The authors should mention if the values correspond to mean and standard deviation.
5. Line 189, Table 3 – Replace “Activities of Days Living” with “Activities of Daily Life”.
6. Line 194 – Replace “HTA=Hypertension Artery” with “AHT=Arterial Hypertension”.
Discussion
7. Line 245 – “average patients” - What do the authors mean?
8. Line 250-251 – “with adequate admitting functional reserves” - What do the authors mean?
9. Lines 256-257 – What about the present study? Did the patients report persistent symptoms even not interfering with daily life activities?
Conclusions
10. Line 314 – I did not understand the use of the word “even” in this sentence.
Reviewer 2 Report
Although its sample size is small, the findings of this study are of great interest in view of the large number of patients who underwent ICU treatment for covid-19.
I have just a few comments.
1. Abstract should mention where study was carried out.
2. All acronyms should be defined when first used (ARDS)
3. A TYPE 1 error of 0.01 conflicts with a significance level of 0.05.
4. Tests for normality carried out on such small samples are meaningless. I suggest to use normal distribution tests/methods throughout without prior testing as these are usually very robust against departures from normality.
Reviewer 3 Report
In my view, the manuscript by Ceruti et al. is well and comprehensibly written and of interest to a wider readership. The aim and design were clearly described. Barthel Index and Karnofsky Index are very basic instruments, but absolutely important and focused on the essential factors (ADLs). Although other studies on EQ-5D, SF-36, PHQ-9 or similar already exist, a comparison between self-perceived health/anxiety/fatigue etc. and the ADLs would have been quite interesting. Is there data on the ability to work of the patients in this study? Limitations are clearly described. However, with a case number of 25 patients, I would point out that the results of the study are only hypothesizing.
Minor points are:
Line 37: SARS-CoV-2 instead of SARS-COV-2.
Figure 1: Arrows seem to be missing from the figure?
Table 1: Please explain the abbreviations below the table. In table 3 it is done. Here you could also refer to the abbreviation explanation in Table 1.
Thank you for the encouraging data!
Reviewer 4 Report
Dear authors,
COVID-19 pandemic has lead to an adundance of post-ICU complications and the identification of the long term effects on the daily tasks seems quite important. I read your manuscript with interest.
Abstract: English check / expression check is needed.
Introduction: Well written.
Methods: Well described.
Please add the approval number of the study by the ethical committee and the acquisition of the informed consent (although it is also mentioned at the end of the article).
-Could also information about whether the patients followed a rehabilitation program be retrieved? So that the improvement be identified as a natural course of the disease or as a result of the interventions in parallel.
Results:
-Concerns on the small number of patients in the study (25 patients). Could the number be increased given the common nature of the disease?
-Please add at the bottom of table 1 the explanations of the abbreviations ex HTA.
-Systolic and diastolic arterial pressures were measured while on vasopressors? If further data are not available on that, either comment on it or remove it. (Table 1) Also heart rate and temperature could be removed on your decision.
-Biological data might need to change to laboratory data (Table 1)
-On secondary outcomes, SOFA shows a statistical correlation with BI at discharge as it is shown in table 3. Please comment on it and change lines 186-187.
Conclusions: Are there similar data on non-COVID patients? How do they compare? (could be added in section 292-298)
Please add a paragraph highlighting the novelty or the additional knowledge added or the cotribution of your article in the literature.
Overall: Although an interesting subject and a well-written manuscript, i have concerns on the small number of patients, on the lack of probably important data as the 1 year routine of the patients, their previous status, the comparison with non-COVID patients (not head to head but at least regarding literature) and the novelty of the study.
Best regards.
Round 2
Reviewer 4 Report
Dear authors,
I find the changed manuscript satisfactory.
Best regards.
Author Response
Thank you.